# Visible and Near-Infrared Hyperspectral Diurnal Variation Calibration for Corn Phenotyping Using Remote Sensing

**Jinnuo Zhang** , **Dongdong Ma** , **Xing Wei** and **Jian Jin** *

Department of Agricultural and Biological Engineering, Purdue University, West Lafayette, IN 47907, USA; zhan4717@purdue.edu (J.Z.); dongdongma812@gmail.com (D.M.); wei408@purdue.edu (X.W.)
* Correspondence: jinjian@purdue.edu

**Abstract:** Remote sensing coupled with hyperspectral technology has become increasingly popular to investigate plant traits, showcasing its advantages in studying plant growth, health, and productivity. The quality of the collected hyperspectral images is crucial for subsequent data analysis and plant phenotyping studies. However, diurnal variations in spectral characteristics introduce more data variance in canopy reflectance spectra, raising the cost of subsequent analyses and compromising the performance of trait estimation models. In this study, a fixed gantry platform in a cornfield was used to capture visible and near-infrared (VNIR) hyperspectral images of corn canopies at consecutive time intervals. By applying reference board calibration and locally weighted scatterplot smoothing to minimize the effects of ambient light and daily growth, diurnal spectral changes across all involved VNIR wavelengths were investigated. Several distinct diurnal patterns were observed to have close connections with the plants' physiological effects. Diurnal calibration models were established at every wavelength by employing the least squares polynomial algorithm, with the highest coefficient of determination reaching 0.84. Moreover, by employing diurnal calibration in canopy spectra processing, the reduction in spectral variance brought about by varying imaging time was evidently exhibited. This study not only reveals the diurnal spectral variation pattern at VNIR bands but also offers a reliable, straightforward, and low-cost approach to improve the quality of remote sensing data and reduce the inherent variance brought about via the different imaging times ensuring that comparable spectral analysis can be performed under relatively fair conditions.

**Keywords:** diurnal spectral pattern; remote sensing; hyperspectral imaging; noise calibration; visible and near infrared

## 1. Introduction

Remote sensing, combined with spectral imaging technology, is becoming an increasingly important and promising tool in crop and agricultural management. It allows for the efficient gathering of critical information that can help optimize crop yield, maximize farmers' profit, and accelerate the progress of plant breeding [1–3]. Specifically, with cost-friendly and high-throughput capabilities, unmanned aerial vehicle (UAV)-based remote sensing is particularly effective in monitoring crop health, nutrient status, soil moisture, and other important traits that can impact crop growth and development [4]. As a data-driven technology, the quality of obtained data in remote sensing plays an essential role in outputting the final decision. However, both the atmospheric effects and instrumental noises can degrade the received hyperspectral radiance [5,6]. In the spectral domain, there are studies that have paid attention to reducing the influence caused by certain signal noises [6]. For instance, blackboard and whiteboard calibration are utilized to maintain the consistency of spectral reflectance under different lighting conditions. In practical applications, it is challenging to eliminate signal noises relying solely on reference board calibration under a constantly changing environment.

Diurnal spectral variation caused by variation in the solar radiation and atmospheric condition also introduces huge signal variance in spectra [7]. Describing and modeling

this diurnal spectral pattern is essential for accurate remote sensing and other applications based on spectral information. Oliveira et al. (2014) discovered that the fluctuation in corn's daily growth cycle had a significant effect on the precision of determining the appropriate nitrogen fertilizer dosage [8]. According to Maresma et al. (2020), the accuracy of yield prediction could also be affected by the spectral index obtained at various times of the day [9]. The diurnal spectral variation was reported in the evaluation of winter wheat biomass as well [10]. Several active and passive regulations, including photosynthesis, transpiration, and stomatal conductance, could be related to the plants' diurnal spectral response [11–13]. Therefore, the spectral reflectance of the same plant has different properties over the course of a day. Without alleviating the diurnal pattern effect, it is challenging to determine whether the variance in the collected spectrum can be attributed to a major treatment difference or simply to the diurnal variance. To prevent this issue, UAV-based spectral image collection is typically scheduled during the sunny daytime to ensure sufficient sunlight for high-quality images. However, it might be difficult to completely avoid the influence of diurnal spectral changes, especially if the entire image capture process takes several hours. Furthermore, there is no standardized timetable for image collection, and the specific timing of these flights will depend on various factors such as the type of crop, the growth stage of the plants, weather conditions, and other variables [14–17]. Hence, it is imperative to construct a model that can effectively calibrate the diurnal variability in order to ensure accurate and reliable measurements. Modeling diurnal spectral patterns at different times of day has the great potential to provide guidelines for remote sensing spectral data quality improvement.

The primary requirement for accurately describing diurnal spectral patterns is to obtain sufficient and accurate spectral features of the crop targets that reflect the actual changes in their spectra throughout the day. Radiative transfer models, such as PROSAIL, are used to understand and simulate the interaction of vegetation canopies' electromagnetic radiation with the environment [18]. These models take environmental factors into account when interpreting the spectral properties of the crop canopy, which manifests the advantages of mathematical calibration. However, these models may not be able to fully capture the impact of diurnal effects brought about by various environmental factors that can change throughout a single day. Luckily, a fixed imaging gantry in the field of Purdue University was able to simulate UAV to collect real-time remote sensing hyperspectral images. Additionally, the diurnal pattern of the corn canopy was partially revealed and parameterized in a former study [19] by using the normalized deviation vegetation index (NDVI). Only two spectral bands in the red and near-infrared range were included. In this study, we further expanded the exploration spectral range for diurnal pattern calibration to all the available wavelengths. The hyperspectral images of the corn canopy were obtained in consecutive measurements with a short time interval at the different growth stages. The goal of this study is presented as follows: (1) to explore the diurnal spectral variation patterns of different wavelengths in the range of visible and near-infrared; (2) to build calibration models for all relevant wavelengths to alleviate diurnal spectral variation at a certain time; and (3) to verify the effectiveness of calibration models through variance comparison.

## 2. Materials and Methods

### 2.1. Experimental Setup

The experiment involved two genotypes of corn, namely P1105AM and B73 × Mo17, which were grown in the open field at the Purdue University Agronomy Center for Research and Education (ACRE) in Indiana, USA. The corn plants of different genotypes were cultivated parallel to each other alongside the gantry imaging tower. A protection crop row was sown closest to the gantry to minimize any additional environmental impact on the experimental plots. Each small plot was of size 3 m and had 15 corn plants. A sample of 12 small plots was selected. At the V4 stage, two fertilization treatments were applied to specific corn plots using nitrogen solutions with different concentrations, respectively. The

detailed information can be found in [19]. The flowchart of the work outlined in this paper is presented in Figure 1.

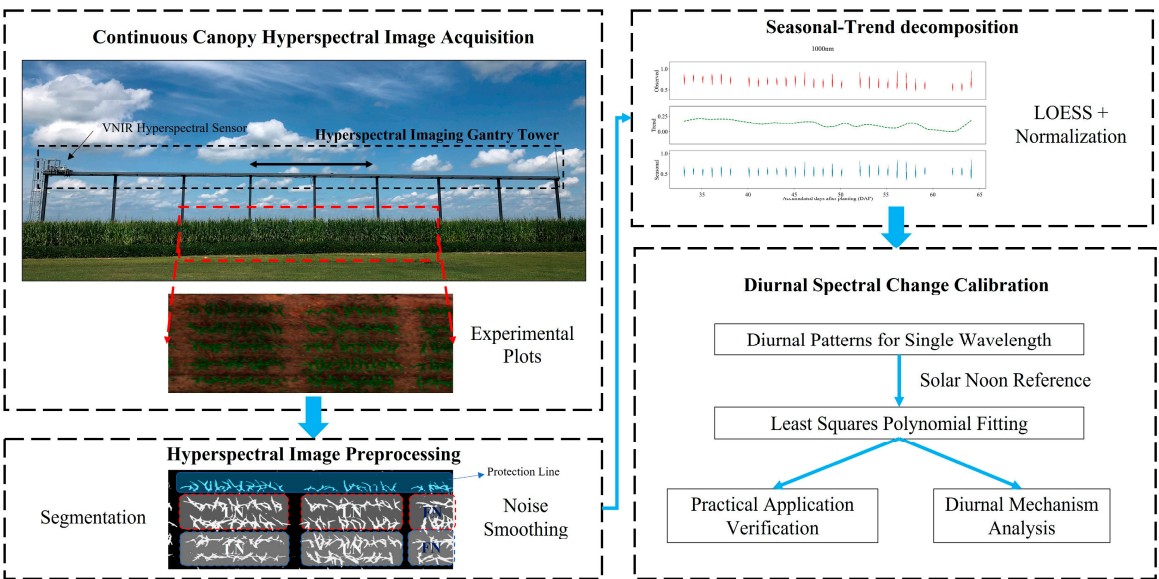

**Figure 1.** The hyperspectral image data collection procedures and the following diurnal calibration model establishment flowchart (FN: full nitrogen treatment; LN: low nitrogen treatment; blue boxes refer to B73 × Mo17 and red boxes refer to P1105AM).

The corn canopy remote reflectance measurements were carried out using a visible and near-infrared (VNIR) hyperspectral sensor (MSV-101-W, Middleton Spectral Vision, Middleton, WI, USA) mounted on the gantry platform. The sensor was positioned 7 m above the ground, and its field of view covered a strip of land measuring 250 square meters. Using the gantry imaging system within a spectral resolution of 1.22 nm, the hyperspectral images of the corn canopy were captured every 2.5 min within the spectral range of 376–1044 nm. This provided the capability for the high-frequency time-series monitoring of diurnal spectral changes. In this experiment, hyperspectral images were acquired for 33 consecutive days after implementing the nutrient treatments. Each day, the gantry imaging system moved backward and forward from 8 am to 7 pm in order to capture the targeted corn canopy reflectance. In total, 8712 hyperspectral images were collected and formed the sample dataset for further processing and modeling. More detailed information about the gantry system can be found in [20].

### 2.2. Hyperspectral Image Preprocessing

To account for variations in illumination conditions and atmospheric effects that could impact the spectral signatures of targets in the scene, a reflectance calibration Formula (1) was applied to all hyperspectral images as the first pre-processing step to correct for ambient light influence. A reference plate attached to the sensors was used to obtain the white reference data, while the reflectance of a black board served as the dark reference during the reflectance calibration process.

$$R = \frac{R_{raw} - R_{dark}}{R_{white} - R_{dark}} \tag{1}$$

where the $R_{raw}$, $R_{dark}$, and $R_{white}$ represent the reflectance of the raw image, dark reference, and white reference, respectively.

The region of interest (ROI) segmentation algorithm, which is based on the distinction between the background field and foreground plants, was employed to extract the average canopy spectra of each individual corn plot. To improve the accuracy of segmentation

results, the classic spectral index NDVI was selected to calculate the heatmap for threshold segmentation. After collecting the average spectra from multiple time points and plots, various spectral data preprocessing techniques were employed to eliminate noise and improve the signal-to-noise ratio, including discrete wavelet transformation, Savitzky–Golay smoothing and moving average smoothing. The discrete wavelet transformation is a powerful tool that allows the resolution of spectral features into multilevel components, representing both high- and low-frequency information [21,22]. The wavelet named 'db9' was utilized with a decomposition level of 5 and a threshold equivalent to 80% of its relative maximum value to suppress high-frequency noise signals. Then, the Savitzky–Golay smoothing, which used a two-degree polynomial function to estimate the central point values of a given window size of 20, was applied. The algorithm eliminated outliers by fitting a curve that followed the overall trend of the spectral data, while still preserving the underlying patterns and characteristics of the data. Additionally, the moving average smoothing, which combined the values in the window size of five, made sure that high frequency noise was removed to the greatest extent possible. Several wavelengths from the beginning and end of the spectra were also removed to mitigate the impact of noise and spectra artifacts brought about by the instrument, leaving a total of 540 wavelengths to be preprocessed. The available spectral data underwent a data quality check based on the interquartile range from a single wavelength, as previously described in [3]. Ultimately, the time interval selected for diurnal analysis was from 10 a.m. to 5 p.m. Eastern Daylight Time, in order to simulate the typical working hours of the practical UAV flights. Together, these hyperspectral image pre-processing procedures helped to refine the spectral data and improve their quality for further analysis.

During the almost month-long experiment of collecting the hyperspectral information of the corn canopy, the natural growth of the plants may have contributed to the inherent variance observed in the spectra. To reveal a clearer diurnal pattern for each individual day, it was necessary to eliminate the influence of daily growth. Therefore, the non-parametric regression method known as locally weighted scatterplot smoothing (LOESS) was selected to decompose the seasonal and trend signals from all the wavelength scatterplots [23]. The VNIR reflectance from various growth stages and genotypes was divided into two major components, representing the growth pattern and the internal variance. To alleviate the growth stage variance, the trend component was subtracted from the original spectral reflectance pattern after a signal normalization.

### 2.3. Diurnal Data Analysis

In this paper, the daily time interval was selected as the primary factor for establishing connections with the diurnal reflectance variations across various wavelengths. First of all, the format of the recording time was converted into decimals for the later establishment of a calibration model. Reflectance values at the same wavelength from different dates were integrated and split into a training set and a testing set at a ratio of 7:3. In each dataset, the reflectance values were averaged based on the daily time data in order to combine multiple outputs. The least squares polynomial curve fitting algorithm, which had a rapid inference ability, was utilized to construct the diurnal calibration models for the 540 wavelengths at different daily time intervals. This algorithm was able to find the coefficients of the polynomial to fit the diurnal pattern by minimizing the sum of the squares of the differences between the reflectance points and the polynomial curve [24]. To prevent severe overfitting, the degree of the fitting polynomial was set to 4 using 10-fold cross-validation. The curve function formula (Formula (2)) and the relative loss formula (Formula (3)) are shown as follows:

$$y = a_0 + a_1 x + a_2 x^2 + \ldots + a_k x^k \tag{2}$$

$$L = \sum_{i=1}^{n} \left[ Y_i - \left( a_0 + a_1 x + a_2 x^2 + \ldots + a_k x^k \right) \right]^2 \tag{3}$$

where $x$ represents the daily time and $y$ and $Y_i$ denote the output reflectance of least squares polynomial curve and the ground truth reflectance, respectively. $a_k$ represents the coefficient at the $k$th polynomial.

On the basis of the learned polynomial coefficient, the diurnal spectral variation pattern at all relevant wavelengths served as a reference table for latter calibration. The ratio matrix of reflectance, derived from two arbitrary time points, represented the relationship between reflectance values and the varying time. Formula (4) illustrated the specific process of this transformation.

$$R_t = \left[ \frac{dR_{t1}}{dR_{r1}}, \frac{dR_{t2}}{dR_{r2}}, \frac{dR_{t3}}{dR_{r3}}, \ldots, \frac{dR_{tn}}{dR_{rn}} \right] \odot R_r \tag{4}$$

where the $dR_{tn}$ represents the reflectance derived from the diurnal curve at the $n$th wavelength and the target time point. $dR_{rn}$ indicates the reflectance derived from the diurnal curve at the $n$th wavelength and the reference time point. $R_r$ and $R_t$ denote the actual reflectance derived from the reference time point and target time point.

*2.4. Evaluation Metrics and Relative Tools*

To assess the performance of the established polynomial models fitted to various diurnal patterns at different wavelengths, the coefficient of determination ($R^2$), which provided the values of the explained diurnal variance and root mean square error (RMSE), was selected as the evaluation metric. Additionally, the standard deviation for each wavelength at varying time points was computed to partially represent the impact of diurnal fluctuations. This analysis offered further insight into the variability of the data and helped to better understand the extent of diurnal influence on each wavelength. The hyperspectral imaging processing algorithms were executed on a Windows 10 operating system powered by an AMD Ryzen 7 5800H CPU. To build and evaluate the diurnal models, open-source Python 3.10.8 (https://www.python.org/, accessed on 6 January 2023) was employed in conjunction with OpenCV and various other public libraries (NumPy, Pandas and so on). This software stack enabled the effective analysis and modeling of the diurnal patterns in the data.

$$R^2 = 1 - \frac{\sum_{i=1}^{i=m}(y_i - \hat{y}_i)^2}{\sum_{i=1}^{i=m}\left(y_i - \bar{y}\right)^2} \tag{5}$$

$$RMSE = \sqrt{\frac{1}{m}\sum_{i=1}^{m}(y_i - \hat{y}_i)^2} \tag{6}$$

where $y_i$ and $\hat{y}_i$ represent the true value and predicted value of sample $i$, respectively. The $\bar{y}$ denotes the averaged value of all samples ($i$ = 1, 2, 3, 4, ..., $m$).

## 3. Results

*3.1. Daily Growth Detrending Results after Applying LOESS*

As depicted in Figure 2A, prior to the decomposition calculation, there was a noticeable upward trend in the NDVI values, indicating an increase in vegetation content [25,26]. However, after applying the LOESS method, the upward trend in the NDVI values was reduced and brought to a relatively consistent level, demonstrating the effectiveness of the detrending process. It is worth noting that the majority of the inter-date neighborhood relationships were well preserved, even after applying the LOESS method. This suggests that the detrending process effectively removed the growth stage variance while still maintaining the overall spectral patterns and relationships between different time points. The detrending algorithm allowed for a clearer diurnal pattern to be revealed for each individual day, facilitating a more accurate analysis of the spectral data.

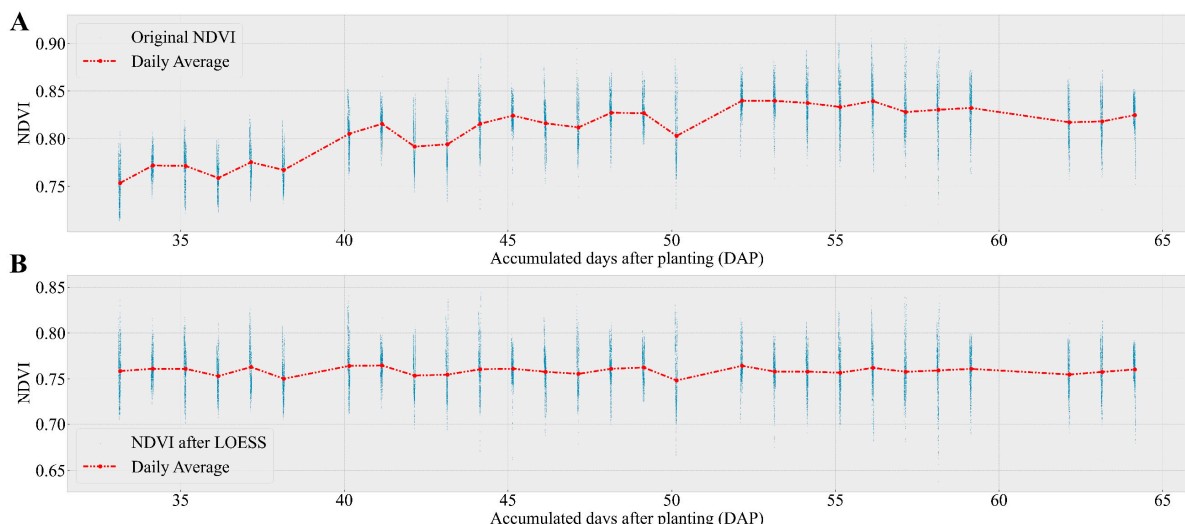

**Figure 2.** The comparison curves result from the original NDVI value at different DAP and the NDVI value after trend and seasonal decomposition based on LOESS. ((**A**): the original NDVI value had relative severe fluctuation with the changing of time; (**B**): the NDVI value after LOESS kept a relative stable fluctuation at different DAP).

### 3.2. Variability in Diurnal Patterns across Wavelengths

Specific wavelengths across whole wavelengths were singled out in order to illustrate their distinct diurnal patterns. As shown in Figure 3, there were nine calibration curves of corresponding spectral bands, which were distributed throughout the entire spectral range. At around 400 nm (Figure 3A), the diurnal pattern exhibited a V-shape with varying time, where the lowest reflectance occurred during the solar noon period. Under this circumstance, the testing phase also achieved an $R^2$ value of 0.84, indicating a satisfying modeling performance. As the wavelength increased (as shown in Figure 3B,C), the V-shaped pattern gradually shifted towards the opposite direction, with the reflectance at solar noon becoming the highest. During this progression, the calibration accuracy of the testing set decreased too. However, when the reverse V-shaped diurnal pattern stabilized around 470 nm, the fitting $R^2$ value increased to a promising level of 0.72. In Figure 3E,F, another stage of the diurnal pattern shifting was displayed, where the reverse V-shaped curves were bent towards the opposite direction again. Eventually, the diurnal pattern within the range of 700 nm to 1016 nm exhibited consistent and uniform V-shaped changes. Almost all of the prediction results within this range exhibited relatively stable and high $R^2$ value, with the exception of the calibration results at 910 nm, which did not perform as well.

In general, it was apparent that there was a notable diurnal variation pattern among various segments in the VNIR range. Figure 4 provided both two-dimensional and three-dimensional visualizations to illustrate the variation surfaces with respect to daily time and wavelength. By applying reflectance normalization, the diurnal pattern within a single day was effectively highlighted. It was evident that within the three sections ranging from 400–470 nm, 470–670 nm, and 670–1000 nm, there were no distinct boundaries or abrupt changes, and the diurnal pattern changed smoothly from a V-shape to a reverse V-shape and vice versa.

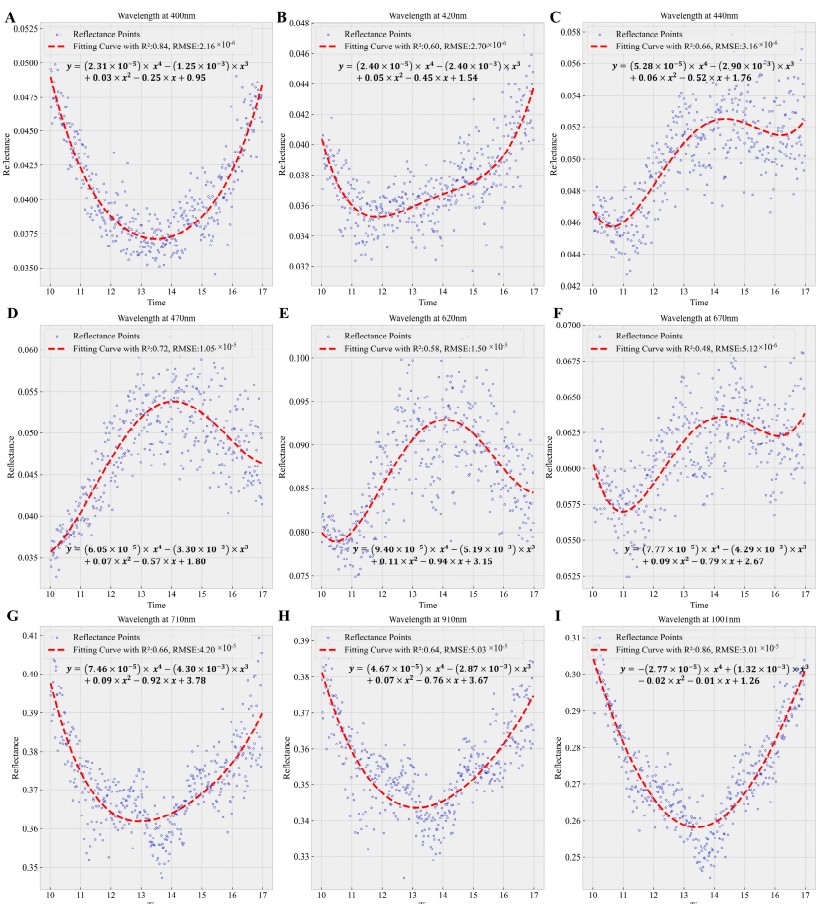

**Figure 3.** Several diurnal calibration curves derived from polynomial models at various characteristic wavelengths ((**A–I**): 400 nm, 420 nm, 440 nm, 470 nm, 620 nm, 670 nm, 710 nm, 910 nm, 1001 nm).

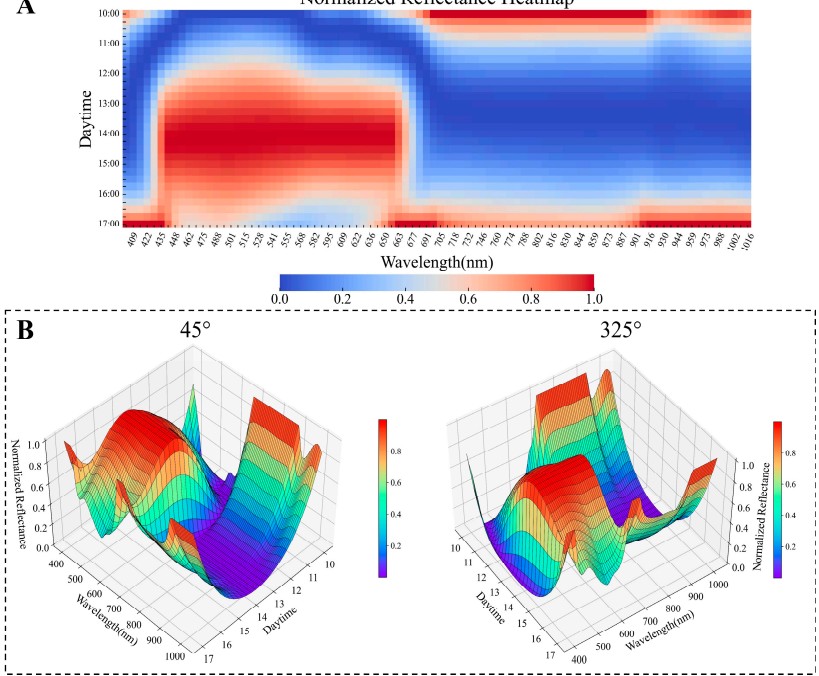

**Figure 4.** The normalized reflectance heatmaps of diurnal changing pattern under varying times of day and wavelengths ((**A**): the view of 2D heatmap; (**B**): the view of 3D surface maps at both 45° and 325° view angles).

### 3.3. Prediction Results from the Diurnal Calibration Model

To further examine the whole picture of these calibration models, considering the daily time as the independent variable, the pattern of diurnal changes at specific wavelengths emerged as the primary factor in constructing calibration models. The complexity of diurnal variation magnitude significantly impacted the models established, emphasizing the importance of understanding these changes to create accurate and reliable models. Figure 5 illustrated the training and testing results of diurnal calibration models utilizing a polynomial with a degree of four. It can be observed that the $R^2$ values in the testing phase were slightly lower than those in the training phase, showing no sign of severe overfitting. Additionally, the relatively low values of the RMSE across all available wavelengths presented the high level of prediction accuracy of the established diurnal calibration model. Different responses occurred during the modeling of wavelength–time-related function. For example, the predictive ability of the diurnal calibration model reached a promising level at approximately 400 nm, 470 nm, 800 nm, and 1001 nm with the $R^2$ values above 0.7. There were also some curve valleys representing relatively low regression performance at several wavelengths, including 420 nm, 670 nm, and 910 nm. These $R^2$ curve dips indicate the areas where the model may struggle to accurately capture the underlying patterns. Overall, the majority of diurnal variances within the investigated daily time could be acquired by the established calibration model.

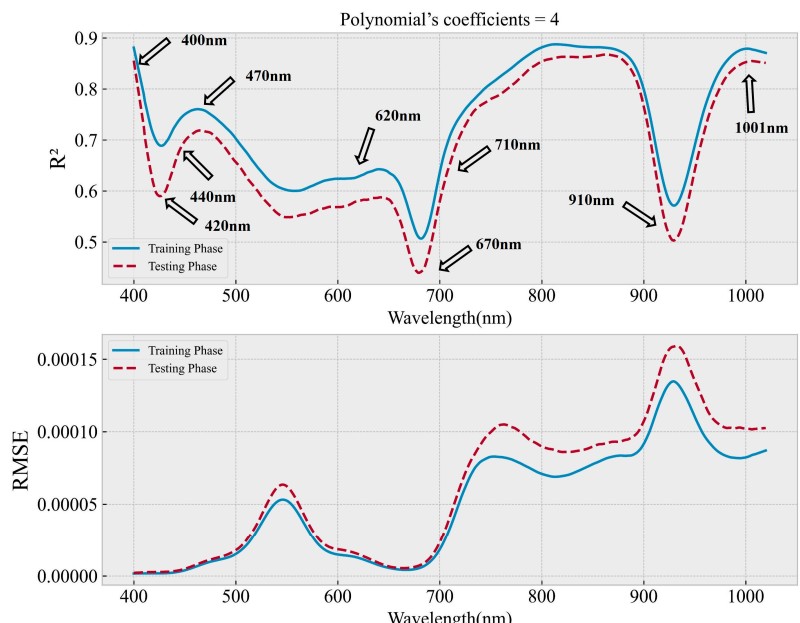

**Figure 5.** The regression performance of the established diurnal calibration model in the VNIR spectral range.

### 3.4. Effectiveness of Diurnal Calibration in Practical Application

Based on the fact that this experiment involved two treatments—full and low nitrogen—the effectiveness of diurnal calibration was further verified by assessing whether the differences between the treatments were maintained while reducing the noise factor introduced via imaging time. Figure 6 illustrates the intuitive results obtained by calibrating the reflectance values of the testing dataset to their corresponding solar noon. In both the P1105AM and the B37 × Mo17 corn hybrids, it was observed that the averaged spectral reflectance derived from the low nitrogen treatment was higher than that derived from the full nitrogen treatment in the visible range, which suggested a decrease in visible light absorbance, indicating a decline in photosynthesis activity (Figure 6A,C) [27,28]. Differences between the two treatments could also be identified in the near-infrared range, where the lower reflectance value of the low nitrogen treatment suggests the possibility of leaf structure damage caused by nutrient deficiency [29,30]. The shaded regions surrounding

these average spectra, which indicated the dispersion of spectral data, overlapped significantly and covered a relatively large area. However, after diurnal spectral calibration, the variation in reflectance was significantly reduced across all wavelengths, as shown in Figure 6B,D. Notably, the spectral difference between the full and low nitrogen treatments was preserved. Reflectance peaks at around 550 nm became even sharper and more distinguishable following the calibration process.

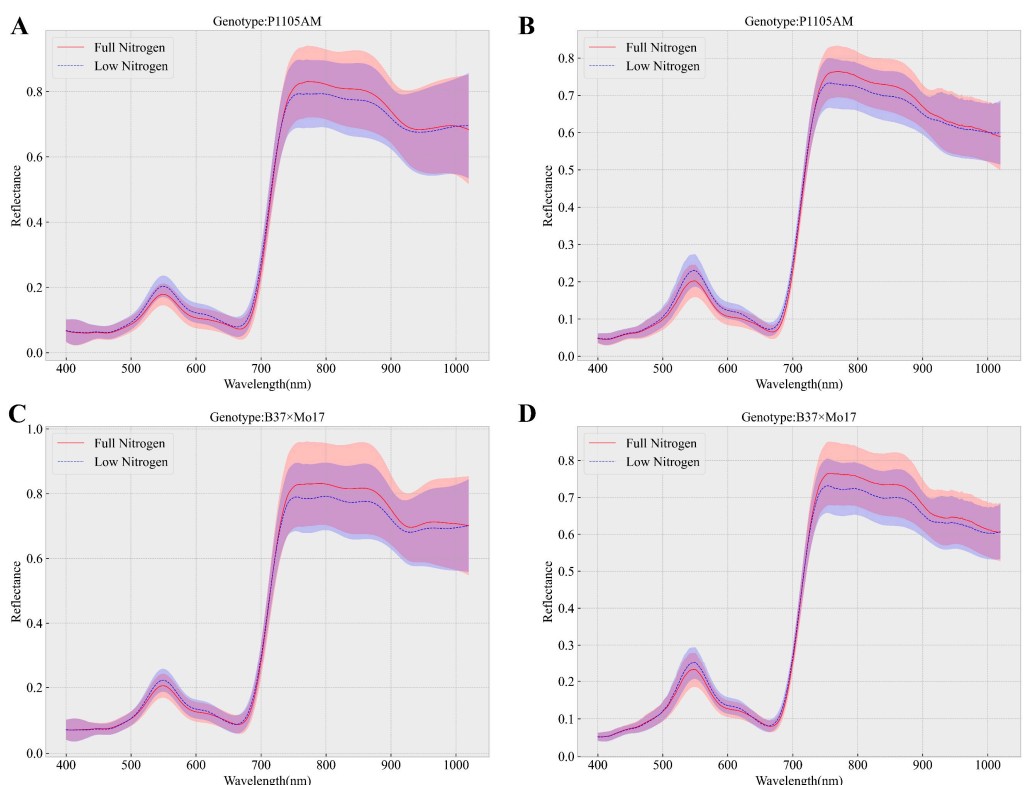

**Figure 6.** The averaged spectra under full nitrogen and low nitrogen treatment from two corn genotypes' testing sets. ((**A**,**C**): the original spectra; (**B**,**D**): the spectra after diurnal calibration to solar noon). The shaded areas indicate the standard deviation range.

From a statistical analysis standpoint, the variance reduction ratio provided detailed information on the effectiveness of the diurnal calibration process. A closer examination of Figure 7A reveals that the reflectance variances at all available wavelengths were reduced, with the overall reduction trend being highly similar to the diurnal model's inference performance. For instance, the range between 800 nm and 900 nm showed the highest variance reduction ratio of 60%. Conversely, when the diurnal calibration model had the least regression $R^2$, around 670 nm, the calibrated variance ratio only reached approximately 28%. Figure 7B elaborated the comparison of the results from the above-mentioned nine wavelengths. In addition to significantly narrowing the range of all variance bars, the relative reflectance values were shifted to different locations via diurnal calibration. This phenomenon may be attributed to the relationship between the reflectance at solar noon and other times of the day. Moreover, it was observed that the number of outliers at several wavelengths decreased compared to the number of outliers in the original data. The diurnal calibration model was found to significantly reduce the noise introduced by imaging time, as demonstrated by the variance reduction observed in the analysis. This highlighted the model's effectiveness in improving the quality of hyperspectral data in remote sensing applications.

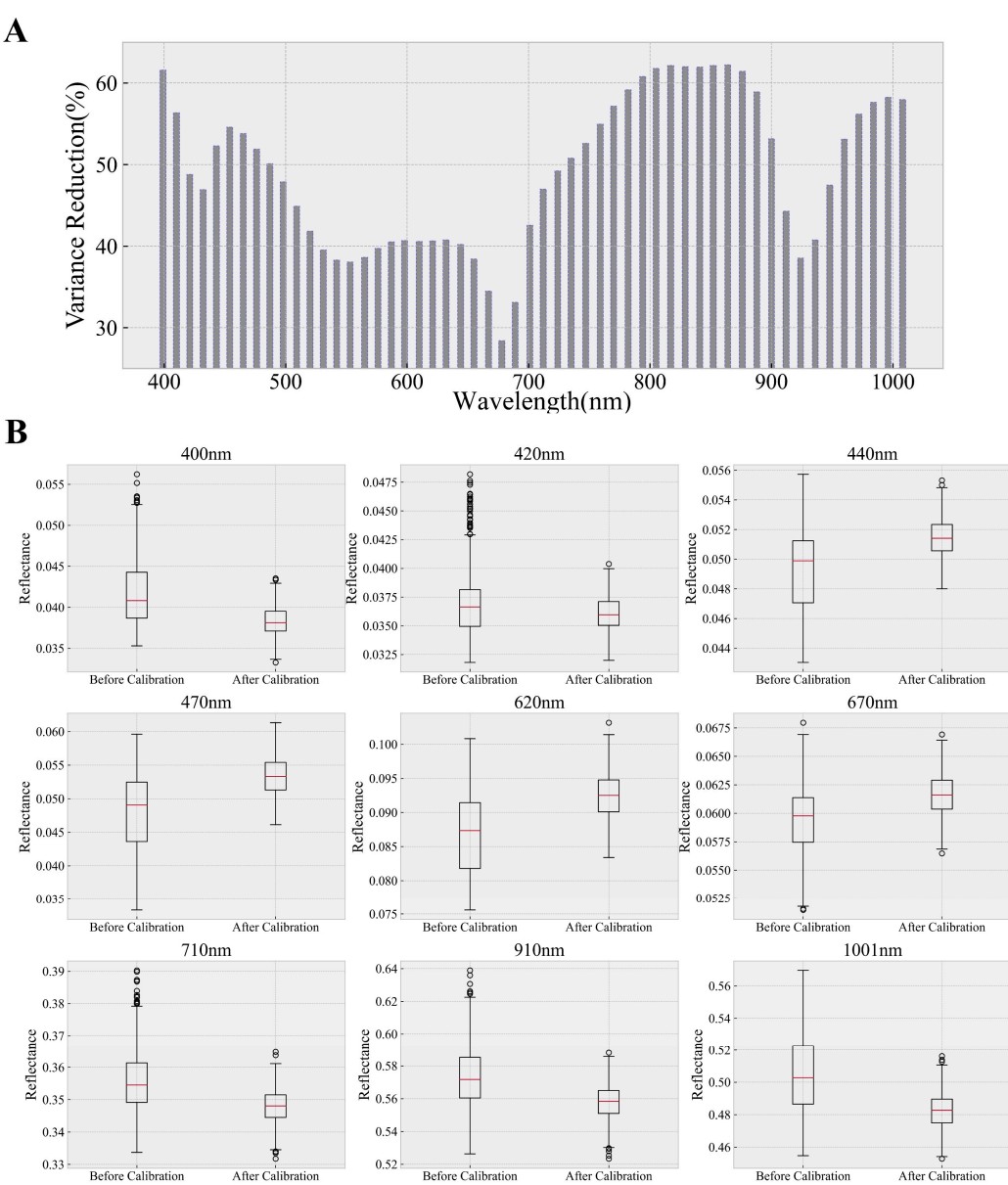

**Figure 7.** Variance reduction ratio at different wavelengths after diurnal calibration ((**A**): variance reduction ratio at all available wavelengths; (**B**): statistical comparison between the before and after diurnal calibration based on reflectance at representative wavelengths).

## 4. Discussion

### 4.1. The Varying Performance of Diurnal Spectral Calibration Models

Compared to Ma's research [19], which concentrated on modeling the diurnal fluctuations in NDVI, this study broadened the calibration scope to encompass the full VNIR spectrum. Throughout this process, more complex diurnal patterns across different wavelengths were uncovered and modeled. By analyzing the results from both Figures 3 and 4, it can be observed that the relatively high predicted $R^2$ values were associated with the stable status of the diurnal spectral pattern. Additionally, the transition phase of diurnal pattern variation, such as at around 420 nm and 670 nm, tended to be associated with lower predicted $R^2$ values. The impact of growth stages and other environmental factors might be relevant in determining the diurnal variations within those spectral ranges [20]. Figure 8A demonstrates that the diurnal patterns at 670 nm exhibited three noticeable differences on various sampling dates, as highlighted by the red boxes. Additionally, it was difficult to identify a consistent pattern of diurnal variation, especially at the time around 16 pm.

At 1001 nm, where accurate predictions were achieved, the overall pattern of diurnal spectral variation was easily discernible, with only one area showing slight divergence (see Figure 8B). Although the overall diurnal pattern retained a consistent U-shape, the prediction performance for the diurnal pattern around 930 nm still decreased. In Figure 9, it can be seen that the reflectance at 930 nm was concentrated around the mean, whereas the reflectance at 960 nm and 1001 nm was more dispersed, indicating greater variance in the data and a more pronounced U-shape. All in all, the varying performance could be attributed to the complexity of the data, which was reflected in both the curve tendency and the variance of the data. As the complexity of the diurnal pattern increases in those wavelengths in the transition phase, modeling it becomes increasingly challenging [31].

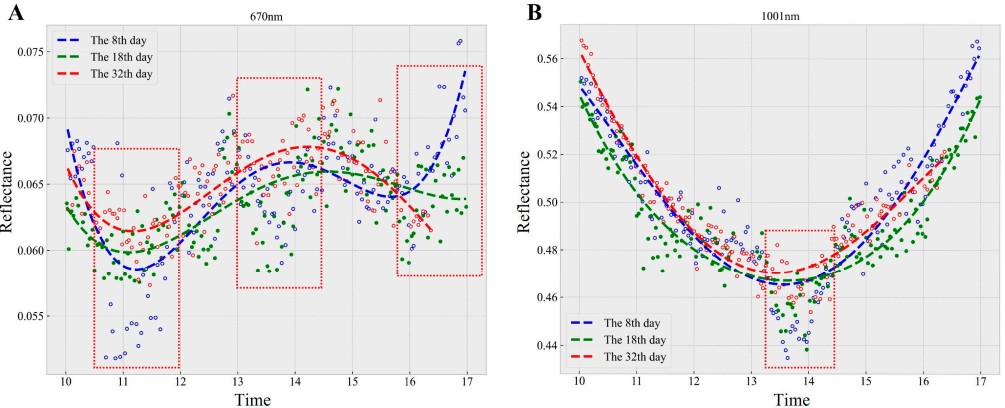

**Figure 8.** Variation in spectral signatures at 670 nm (**A**) and 1001 nm (**B**) during diurnal cycles across diverse growth phases.

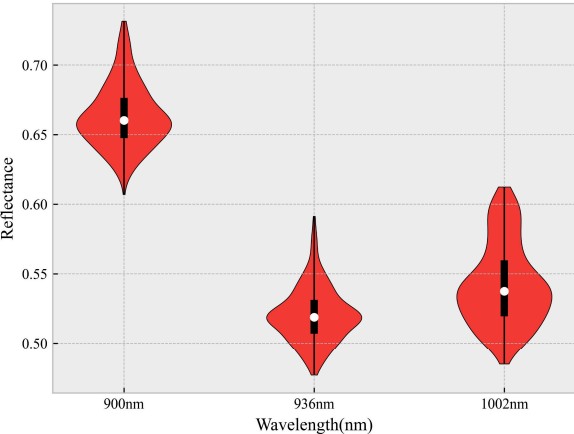

**Figure 9.** Statistical plot of the reflectance distribution at 900 nm, 936 nm, and 1002 nm.

### 4.2. The Mechanisms of Diurnal Spectral Variation Patterns in VNIR

In this work, the imaging time was treated as the primary independent variable to predict the diurnal spectral variation. The hidden mechanism causing this time-related variation could be attributed to many factors, such as the solar radiation, atmospheric conditions, environmental influences, the self-regulation of plant, and so on [32]. Usually, these factors would interact with each other and affect the normal status of plants, leading to the ultimate spectral variation in a combined manner. Considering the characteristics of reflectance spectra, which were highly dependent on incident light, solar light intensity could be the dominant factor influencing the spectral response of crops at different times throughout the day. As the day progresses, the solar altitude angle initially increases, reaching its peak at the solar zenith angle (when the sun is directly overhead) and then gradually decreases until sunset. In addition, it was previously reported that the intensity

of light follows the same pattern to interact with the corn plants, resulting in a spectral alteration. Milton et al. have found that there was connection between the solar zenith angles and the performance of the crop trait estimation based on spectral indices [33].

Although the consistent pattern of solar radiance intensity played a role in shaping the diurnal patterns, it did not fully account for all the variations observed in Figure 4, where distinct diurnal spectral patterns were seen at different wavelengths. For instance, the spectral features between 400 nm and 500 nm displayed a shift from a clear diurnal V-shape to a reversed diurnal V-shape. Given that the violet-blue light in the range of 400–450 nm is predominantly absorbed by chlorophyll a and b, the observed diurnal V-shape may reflect changes in photosynthetic activity [34]. With the decrease in light absorption and the increase in light intensity, the diurnal spectral pattern gradually became a reversed V-shape. Furthermore, the diurnal curves in the range between 500 nm to 600 nm, which represent the green light, further supported the hypothesis that as the solar radiance increased, the amount of light reflected by the corn leaves also increased, leading to the formation of the reversed V-shape pattern. As the chlorophyll and other pigments inside the corn leaf regained the light absorption ability in the spectral range (600–700 nm), the diurnal spectral pattern shifted into a V-shape again [35]. In the near-infrared range, the spectral reflectance could be an indicator of the water status of the plant leaves [2,36]. The low reflectance at solar noon in these spectral wavelengths could be related to the plant self-regulation of transpiration and stomatal conductance. It has been demonstrated in the previous study that the relative water content of the corn crops peaked approximately at solar noon [19]. Consequently, as the water content increased, the spectral reflectance values tended to decrease.

### 4.3. The Application and Prospective of Diurnal Spectral Calibration

Leveraging the strong inferential capabilities of the well-established diurnal spectral calibration model, the time-dependent noise presented in remote sensing hyperspectral data could be significantly minimized. This is the first time that diurnal spectral variation was systemically quantified and parameterized. Regarding practical applications, the researchers were able to arbitrarily transform their data to a random but uniform time phase, subsequently improving the raw spectral signal quality as well as alleviating the influence of ambient light. In addition, the identified diurnal spectral patterns might also have the potential to guide the design of UAV-based remote sensing experiments, optimize the appropriate timeline for data collection, and reduce the impact of the data source to the maximum extent possible. However, there are still some areas of improvement in this study that could be addressed in future research. Firstly, incorporating a broader range of crop fields with diverse plants cultivated in different locations and over multiple years would contribute to the development of a more robust diurnal spectral calibration model. Secondly, it is essential to utilize advanced feature extraction and analysis algorithms, such as deep learning, to more accurately decipher the intricate information embedded in diurnal spectral patterns. Additionally, a thorough investigation of the underlying mechanisms within the diurnal spectral patterns across the visible and near-infrared (VNIR) spectral range is necessary, as it would offer more evidence and tools for mitigating the impact of complex noise sources.

### 5. Conclusions

The diurnal variations in the VNIR spectra of crops captured by UAV-based remote sensing can introduce serious temporal noise, which may inevitably affect data analysis and modeling efforts. In this work, the VNIR spectra of a corn canopy in an open field across different growth stages was obtained every 2.5 min using a fixed gantry imaging tower to study the diurnal patterns of spectral variation. Not only was the diurnal tendency at all available wavelengths uncovered but the essential features were also successfully identified and calibrated using a combination of the least squares polynomial fitting algorithm and the LOESS algorithm. Moreover, the effectiveness of the established diurnal calibration

model was preliminary verified by comparing the reductions in the standard deviations within the spectra data. Overall, the diurnal calibration model proved its potential to clearly reduce the noise brought about by the imaging time in the form of variance reduction, demonstrating its ability to improve remote sensing data quality.

**Author Contributions:** Conceptualization, J.Z. and J.J.; methodology, J.Z. and J.J.; software, J.Z.; validation, J.Z., D.M., X.W. and J.J.; formal analysis, J.Z.; investigation, J.Z.; resources, D.M. and J.J.; data curation, D.M.; writing—original draft preparation, J.Z.; writing—review and editing, J.Z., X.W. and J.J.; visualization, J.Z.; supervision, X.W. and J.J.; project administration, J.J.; funding acquisition, J.J. All authors have read and agreed to the published version of the manuscript.

**Funding:** This research was funded by Purdue University.

**Data Availability Statement:** The data presented in this study are available on request from the corresponding author.

**Acknowledgments:** This work was sponsored by the Department of Agricultural and Biological Engineering at Purdue University.

**Conflicts of Interest:** The authors declare no conflict of interest.

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
