# Peer review of "Visible and Near-Infrared Hyperspectral Diurnal Variation Calibration for Corn Phenotyping Using Remote Sensing"

_remotesensing, doi:10.3390/rs15123057_

Round 1

Reviewer 1 Report

This manuscript, which analyzes the daily variation of visible and near-infrared hyperspectral images of the maize canopy, combines the least squares method to develop a separate daily variation calibration model for each band. The method is useful for ensuring the quality of remote sensing data. However, the authors are not precise in describing the construction of the calibration model and how to apply it.

1. The acquisition time of spectral images is influenced by geographical location, monitoring indicators and other factors, and the monitoring of some parameters (e.g., daily variation of plant photosynthesis and respiration mentioned by the authors in the paper) requires exactly such differences. Also, at the same time, the spectral images of crops under different growth conditions can differ, for which how does the correction model avoid the elimination of these differences? The authors are expected to clarify the significance of the study in the abstract and introduction.

2. I do not quite understand how the authors made the correction. In 3.1 the authors gave the fitted curves for the daily variation of spectral reflectance in different bands and analyzed the distribution of reflectance at different times in different spectral bands. The fitted curves of the polynomial and finally the application of the corrected data is given. I believe that the fitted curves given by the authors in the first two sections can at most determine the spectral reflectance of a band at a certain time, without any correction. I wish the authors would have reflected on how the correction was made in the manuscript.

Abstract

Line12 Whether the authors' NOISE in the manuscript is accurate and the daily variation of the plant itself has an effect on the spectral reflectance, which is important in the monitoring of some parameters, should be called noise.

Line 15. Why reduce the impact of daily growth?

Line 18-22, what specifically is the benefit of calibrating the daily variation of the spectrum and whether the advantage of the corrected spectrum can be quantified?

Introduction

The introduction to this manuscript is weak and does not clearly sort out what has been done and what problems remain in the current relevant research, nor does it clearly summarize the significance of this study.

Line 61-63 For different latitudes and different crops, the best time to acquire the canopy spectrum is different. The author only uses the different collection times of two kinds of literature to illustrate that the research significance of this manuscript is too far-fetched.

Line 67-68 This study does not involve the PROSAIL model

Materials and Methods

Figure 2 and the related contents of the analysis Figure 2 are more appropriately placed in Results and Analysis.

Line 190, whether R2 is used as the evaluation can explain the problem, it is suggested to add the root mean square error and other indicators for joint evaluation.

Results

What is shown in Figure 3, and I hope the authors can explain why fitting the daily variation of the spectrum of a certain band, and what is the role of the magnitude of its R-squared? Also, it is suggested to add the fitted curve equation.

Figure 5. The trend of R² in different bands is represented by a continuous curve. First of all, the authors only used 9 bands to fit the daily variation and could not determine whether there would be fluctuations between different bands, and the authors did not explain why they used a quadratic polynomial for fitting, so I think it is not reasonable to use a continuous curve and suggest a bar graph; in addition, I hope the authors can explain why they used to understand the trend of R² in different bands, what is the significance for the calibration of the daily variation of the spectrum?

Minor editing of the English language required

Reviewer 2 Report

General comments:

The manuscript “Visible and Near-infrared Hyperspectral Diurnal Variation Calibration for Corn Phenotyping Using Remote Sensing” investigated the use of spectral calibration models to quantify and parameterize diurnal spectral variations associated with hyperspectral data for corn fields. In general, it is a well-written manuscript with novel contributions and would be of interest to the audience of Remote Sensing. I have some small suggestions/ minor revision requests to perhaps help the authors further improve their manuscript.

Specific comments:

L37. Explain “reference board calibration” for the broad agriculture and environmental science audience.

L47. Check if year needs to be added for in-text citation.

L81. The ‘daily time’ wording needs to be changed for better clarification.

Figure 1. It’s a bit hard to see “FN” and “LN” in the images.

L127. NDVI has already been spelled out previously.

L138-139. Awkward sentence.

L255-258. It might be helpful to move these to the intro or method to better set up why different treatments are used.

L351-353. Awkward sentence.

L365. Has this been done for remote sensors? Would findings from this work be applicable to sensor studies other than those used UAVs? 

Minor editing is needed for this manuscript. I pointed out some awkward sentences that need to be fixed.

Round 2

Reviewer 1 Report

As the authors say the study is very meaningful for correcting spectra acquired at different times to the same time level, and I think this study is more suitable for proximal sensing.

1.The correction mechanism explained by the authors should eventually be achieved by calibrating the spectra obtained at different times to the same time (noon). I think the correction model should input spectral acquisition time and spectral reflectance and output spectral reflectance at noon. While the correction model currently constructed by the author only seems to find spectral reflectance at different time under different wavelengths. It's not clear how the reflectance at different times relates to the noon reflection coefficient. I think the verification mentioned in 3.4 seems impossible based on the results the authors have given so far in the paper.

2. Please add the formulas in 2.4 to explain how to calculate the “R” and “RMSE”.

没有评论
